# Implications of Dosage Deficiencies in CTCF and Cohesin on Genome Organization, Gene Expression, and Human Neurodevelopment

**DOI:** 10.3390/genes13040583

**Published:** 2022-03-25

**Authors:** Christopher T. Cummings, M. Jordan Rowley

**Affiliations:** 1Munroe-Meyer Institute, Department of Genetic Medicine, University of Nebraska Medical Center, Omaha, NE 68198, USA; chris.cummings@unmc.edu; 2 Genetics, Cell Biology and Anatomy, University of Nebraska Medical Center, Omaha, NE 68198, USA

**Keywords:** architecture, chromatin, cohesinopathy, CTCF, dosage, extrusion, Hi-C, looping, mutation, neurodevelopment

## Abstract

Properly organizing DNA within the nucleus is critical to ensure normal downstream nuclear functions. CTCF and cohesin act as major architectural proteins, working in concert to generate thousands of high-intensity chromatin loops. Due to their central role in loop formation, a massive research effort has been dedicated to investigating the mechanism by which CTCF and cohesin create these loops. Recent results lead to questioning the direct impact of CTCF loops on gene expression. Additionally, results of controlled depletion experiments in cell lines has indicated that genome architecture may be somewhat resistant to incomplete deficiencies in CTCF or cohesin. However, heterozygous human genetic deficiencies in CTCF and cohesin have illustrated the importance of their dosage in genome architecture, cellular processes, animal behavior, and disease phenotypes. Thus, the importance of considering CTCF or cohesin levels is especially made clear by these heterozygous germline variants that characterize genetic syndromes, which are increasingly recognized in clinical practice. Defined primarily by developmental delay and intellectual disability, the phenotypes of CTCF and cohesin deficiency illustrate the importance of architectural proteins particularly in neurodevelopment. We discuss the distinct roles of CTCF and cohesin in forming chromatin loops, highlight the major role that dosage of each protein plays in the amplitude of observed effects on gene expression, and contrast these results to heterozygous mutation phenotypes in murine models and clinical patients. Insights highlighted by this comparison have implications for future research into these newly emerging genetic syndromes.

## 1. Introduction

Genetic material is expertly and intricately stored in three-dimensions to provide the context for transcription regulation. 3D genome folding creates long-range chromatin-chromatin interactions between distant loci, allowing enhancers to contact promoters even when separated by kilobases, or even megabases, of sequence. Studies from the last decade indicate multiple organizational features of mammalian genomes (reviewed in [1]). A prominent layer of chromatin organization is the segregation of active and inactive chromatin into A and B compartments. Another layer are chromatin loops forming enhancer-promoter interactions. As part of these multiple interconnected organizational layer are the multitude of high-intensity chromatin loops characterized by CCTC-Binding Factor (CTCF) and cohesin. CTCF and cohesin have been implicated in a multitude of diverse cellular functions, including DNA repair, sister chromatid cohesion, and the proper localization of DNA methylation and histone modifications (Reviewed in [2,3]). In addition CTCF, as an 11 zinc-finger protein, occupies thousands of loci which act as the anchors of intense chromatin interactions, termed CTCF loops. Largely thanks to sequencing-based approaches like Hi-C, and in conjunction with CRISPR-mediated inversion of motifs, it was shown that the orientation of CTCF binding is important for loop formation [4,5,6]. This orientation preference can be explained by a model in which loops are extruded through cohesin until blocked by properly oriented CTCF protein (Figure 1) as reviewed elsewhere [1,7,8,9,10]. In recent years, the loop extrusion model has been supported by a large amount of evidence, although limitations to the model remain, including an explanation for the ability of cohesin to bypass some CTCF cites while passing others [11,12,13,14,15]. This extrusion process to form CTCF loops is thought to provide the context for interactions between regulatory elements, including enhancers and promoters, and may thereby influence gene transcription [16,17]. However, recent work has led to questioning the impact that these loops have on transcription regulation, as well as how incomplete deficiencies may alter genome architecture [18,19,20]. We review these works below, and contrast these findings to the dramatic in vivo phenotypes when CTCF or cohesin are semi-deficient.

Clinically, disruption of factors involved in forming CTCF loops is becoming increasingly recognized as a cause of human genetic disease [21]. Germline variants in CTCF and in core cohesin complex members have been identified in increasing numbers of patients over the past several years. Characterized in large part by intellectual disability (ID) and developmental delay (DD), these disorders illustrate the critical importance of proper chromatin organization in the developing human brain. Additional anomalies in diverse body systems are also recognized in these disorders, hinting at the role of CTCF looping in development more broadly. Distinct from mutations in the *CTCF* gene, patients with a mutation in a single CTCF binding site have also been identified and show various congenital anomalies, again demonstrating the role of this process during development [22,23,24,25].

Here we compare and contrast the different organizational and functional effects that deficiencies in CTCF and cohesin exhibit in cell line models, animals, and patients. We specifically examine the sensitivity or resiliency to CTCF and cohesin protein levels in each system. Additionally, it is curious that many patients who have deficiencies in these proteins exhibit distinct neurodevelopmental-related phenotypes; therefore, we highlight this neuro-bias and contrast the prevalence of these findings with those involving other body systems. A discussion of these results highlights the importance of architectural protein dosage in disease and the implications this carries for the direction of future studies aiming to understand the underlying mechanisms of these neurodevelopmental disorders.

## 2. Main

### 2.1. Chromatin Organization and Gene Expression upon CTCF or Cohesin Depletion

CTCF was originally identified as a transcriptional regulator [26]. Importantly, removal of the upstream CTCF binding site resulted in dramatic changes to c-myc expression in chickens [26]. Since then, there has been much interest in deciphering the role of CTCF in transcription regulation, particularly as it relates to genome organization. Reporter assays revealed that CTCF can insulate enhancers from promoters, indicative of a role in enhancer-promoter specificity [27,28,29]. The subsequent discovery that CTCF creates thousands of loops throughout the genome [5] was thought to explain this insulator activity, such that enhancers and promoters can be isolated into separate loop domains [16]. Intriguingly, even though CTCF correlates well with insulation and chromatin looping, early efforts at depleting CTCF in cell lines resulted in less than expected changes to chromatin organization. For example, siRNA for 48 h depleted CTCF by 80%, but resulted in surprisingly minor changes to insulation [30]. CTCF depletion to this level also resulted in only 161 differentially expressed genes [30].

Similarly, early studies depleting the cohesin subunit Rad21 resulted in only modest effects. For example, Seitan et al. utilized the Cre recombinase system to delete the RAD21 locus in mouse thymocytes, resulting in decreased protein expression by 70–93% in the cellular population [31]. Hi-C revealed that chromatin organization was largely unaffected, with a few modest changes to individual compartment intervals. As a caveat, Hi-C experiments at that time in general lacked the resolution to look directly at CTCF loops [5]. Regardless of these limits, RNA-seq revealed 1153 statistically significant differential genes; however, most of these were <4 fold changes. Sofueva et al. similarly utilized a Cre-based system to deplete RAD21, in this case by 89% in neural stem cells derived from mouse embryonic stem cells, which were subsequently differentiated into astrocytes [32]. After 96 h Hi-C analysis again showed only modest changes, with domain borders that were overall unchanged, although differences in the frequency of inter-domain and intra-domain contacts were detected, signifying reduced insulation at some level. Examination of gene expression revealed 1762 differential genes, suggesting that only a subset of genes are controlled by cohesin even after an extended depletion period. Zuin et al. used a protease system to deplete RAD21 protein levels by 70–80% within 24 h [30]. Hi-C analysis once again demonstrated only modest effects of reduced interactions between interaction domains, but did not detect significant changes in domain positions or boundaries. In this study, only 48 genes showed differential expression after RAD21 depletion. Intriguingly, Zuin et al. found that RAD21 depletion disrupted the expression of different genes than those affected by depletion of CTCF, which is consistent with their distinct roles in loop formation (Figure 1). Overall, despite significant depletion of CTCF or RAD21, sometimes down to only c.a. 10% the original levels, the effects on genome architecture and gene expression were more subtle than what may be expected of proteins integral to genome organization.

More recent use of the auxin-inducible degron (AID) system achieved acute depletion of architectural proteins to far lower levels than was previously possible. Several labs have independently used the AID system to acutely degrade CTCF to background levels (estimated at <1% remaining) (Figure 2), and demonstrated major changes to genome architecture, with decreased insulation and loss of CTCF loops in as little as 2 h [18,33,34]. Similarly, several labs have used AID to deplete cohesin subunits which resulted in loss of CTCF loops [19,33,35,36].

Despite dramatic changes to CTCF looping, acute depletion of CTCF or cohesin had minimal impact on gene expression with only a few dozen to a few hundred differential genes in each case (Figure 2). Interestingly, Nora et al. demonstrated that longer depletion times lead to more dramatic effects [18]. For example, 370, 1353, and 4996 differential genes are seen after depleting CTCF for 1, 2, or 4 days, respectively. This fits with earlier non-AID studies which saw slightly more than a thousand differential genes, but which required longer depletion times (Figure 2). This also suggests that CTCF looping may be required to maintain transcription programs over long term. Additionally, it is possible that the immediate changes in the expression of a small number of genes lead to secondary effects that culminate in the larger expression changes seen at longer time-points.

While depletion of either CTCF or cohesin results in a loss of original loop signal in these experiments, the mechanistic reason is dramatically different. Deficiencies in cohesin result in loss of defined loops due to the widespread loss of chromatin extrusion. In contrast, deficiencies in CTCF result in loss of extrusion blockage sites, thereby allowing cohesin to form random loops (Figure 1). This likely explains why gene expression defects do not overlap between cohesin v.s. CTCF depletion experiments.

### 2.2. Evidence of Dosage Sensitivity to CTCF and Cohesin Depletion

Contrasting the AID-mediated acute depletion studies to those using other methods can provide insights into the different effects on chromatin architecture and gene expression (Figure 2). As mentioned above, the pronounced gene expression defects of earlier studies may be explained by their longer depletion times. Curiously, the more pronounced effects on gene expression corresponded to more modest changes in chromatin architecture compared to the later experiments using AID, thus the question remains as to why these non-AID studies depleting CTCF or cohesin displayed only modest changes in chromatin architecture.

Interestingly, AID-tagged controls themselves can partially answer this question. Addition of the AID tag to CTCF or cohesin resulted in partial depletion by about half, even in the absence of auxin addition [18,33,35,36]. Despite a 2-fold decrease in protein level, there were minimal differences in chromatin architecture comparing to wild-type cells. In agreement with incomplete knockdown studies, this suggests that genome architecture may be semi-resistant to changes in CTCF or cohesin levels. This point was further demonstrated in the work by Nora et al. in which 5C experiments were repeated with various doses of auxin [18]. This showed that depletion of CTCF to 15% (i.e., 85% depletion), was insufficient to detect major changes in chromatin organization, with the most dramatic changes only appearing after depletion to <1% (Figure 2).

On the surface, these cell-line depletion experiments suggest that chromatin architecture is resistant to architectural protein dosage, and that only near complete loss of the proteins will significantly impact chromatin organization. However, the resiliency of genome organization and gene expression to CTCF and cohesin dosage in these cell-line experiments is difficult to reconcile with the pronounced phenotypes in animal models and in patients that exhibit heterozygous mutations in these genes which we discuss below. Related to this discrepancy, it is important to note the limitations of these cell-line depletion experiments assessing chromatin architecture, for example that techniques such as Hi-C typically do not have the regulation of CTCF loops very highly resolved in a spatiotemporal manner, which may play a role in the observed phenotypes.

Regardless, these findings suggest that the majority of CTCF loops are preserved unless the architectural protein is practically absent, and that only a small subset of CTCF loops were sensitive to more moderate changes in dosage levels of either protein [30,31,32]. It is curious, therefore, that depletion level of 90% is insufficient for loss of CTCF loops, but results in a large number of differentially expressed genes, even accounting for the longer depletion times that were used (Figure 2). Recent work has used CRISPR to insert, delete, or invert CTCF binding sites to alter the local chromatin organization, some reporting changes to nearby gene expression, while others find minimal impact [6,20,38,42,43,44,45]. Therefore, the degree to which individual CTCF loop anchors impact gene expression is likely context specific.

As mentioned above, it is also unclear if these changes in gene expression are actually directly due to the loss of loops. Reporter assays revealed that CTCF from *Drosophila melanogaster* acts as an enhancer-blocking insulator protein that can affect gene expression [28,29]. However, high-resolution Hi-C experiments revealed that CTCF does not anchor high-intensity punctate loops in *Drosophila* [46,47]. This suggests that CTCF’s control of gene expression evolved separately from its role in chromatin looping. It is also important to note that *Drosophila* CTCF interacts with, and binds chromatin alongside, other architectural proteins such as CP190, indicating that its mechanisms may function differently [48,49,50]. Interestingly, recent conditional deletion experiments, revealed that *Drosophila* CTCF is particularly important for proper gene expression in neuronal cell types [29].

### 2.3. Deletion of CTCF and Cohesin in Mouse Models

Murine models have proven useful to evaluate the phenotypic impacts of the loss of mammalian CTCF and cohesin in vivo, and are a critical tool for future research given the reduction of confounding factors when compared to studies focused solely on clinical patients (Figure 3). Homozygous deletions of CTCF, RAD21, SMC3, STAG1, and STAG2 result in early embryonic lethality, starkly revealing an intolerance in vivo of complete depletion of these proteins [51,52,53,54,55,56]. Due to this embryonic lethality, conditional knockouts are more often used to assess the role of each in the development of specific tissues or structures.

For example, Watson et al. generated a CTCF conditional knockout mouse model in subsets of proliferating neuroprogenitor cells at two separate early embryonic time-points [57]. Loss of CTCF resulted in profoundly hypocellular and disorganized brain development, resulting in death at or shortly after birth. Illustrating the role of CTCF in the maintenance and survival of neuroprogenitor cells, they found massive apoptotic cell death and induction of premature neurogenesis of neuroprogenitor cells which depleted the pool of progenitor cells necessary for normal development. Looking at a later stage of neurodevelopment, both Hirayama et al. and Sams et al. have described mouse models with conditional knockout of CTCF in separate subsets of postmitotic neurons [58,59]. Interestingly, although mice were viable at birth in each case, lifespan was shortened, and significant behavioral and learning deficits were demonstrated. Study of hippocampal neurons revealed massive apoptosis, similar to that found by Watson et al. [57]. Depletion of CTCF primarily in the dorsal telencephalon revealed structural anomalies in dendritic arborization and spine density. Together with the findings from *Drosophila* mentioned above, these experiments indicate that CTCF plays a key role during neurodevelopment.

Cohesin’s impact on neurodevelopment has also been illustrated by conditional knockout mouse models. Conditional CRISPR knockout of RAD21 within the anterior dorsal cerebellar vermis of adult mice disrupted their ability to learn a conditioned startle response, demonstrating a defect in associative motor learning [60]. Mice heterozygous for SMC3 developed abnormal brain architecture, and increased anxiety-related behavior, while complete deletion of SMC3 in neurons resulted in growth restriction and early death [55]. Finally, mice heterozygous for STAG1 loss had a shortened lifespan and features consistent with premature ageing [54]. Together, these experiments have clearly implicated cohesin as essential for neurodevelopment.

While we have focused on neurodevelopment, conditional depletions of CTCF and cohesin-complex members have demonstrated their importance in the development of other tissues, including the heart, limbs, immune system, blood cells, and craniofacial structures [55,61,62,63,64,65,66].

### 2.4. Implications of Germline Variants of CTCF and Cohesin in Humans

Perhaps the most convincing evidence for the critical importance of CTCF and cohesin in human brain development is demonstrated by the phenotype of clinical patients found to have germline variants in *CTCF* or in cohesin-complex member genes. Over the past ten years, heterozygous germline variants in CTCF, RAD21, SMC1A, SMC3, STAG1, and STAG2 have been identified in individuals. Interestingly, these patients have predominantly neurodevelopmental phenotypes (Figure 4).

CTCF deficiency was first reported in 2013 in three individuals with the aforementioned core features of intellectual disability (ID), microcephaly, and growth retardation [67]. Additionally, these patients provided early clues toward the importance of CTCF in the development of other tissues, as congenital heart disease, cleft palate, gastrointestinal problems, and genitourinary anomalies which were identified in one or two of the identified patients. Subsequent reports would confirm developmental delay (DD), intellectual disability (ID), and growth restriction as central to the phenotype, while also broadening the phenotypic spectrum by reporting a wide variety of additional congenital anomalies, although at a lower frequency than the core features [68,69,70,71].

A recent report summarized the findings of 39 known individuals, the largest sample size to date, allowing further refinement of the phenotype [72]. DD and/or ID was found to be present in all individuals, suggesting that CTCF levels are particularly important to developmental neurobiology. The wide spectrum of neurocognitive outcomes was notable, including individuals with minimal apparent neurocognitive effects to those with severe ID. This wide spectrum of neurodevelopmental phenotypes is also shown through familial cases where significant intra-family phenotypic variation has been observed [71,72]. Recently, due to large-scale studies focused on candidate gene identification in groups of patients with clinical diagnoses, there has been an increase in individuals identified with variants in CTCF. These have included patients with DD, ID, autism spectrum disorder, critically ill newborn infants, Tourette disorder, and obsessive-compulsive disorder [73,74,75,76,77,78,79,80]. Thus, despite the resistance of architectural changes to dosage and minimal effects on expression found in cell line models, patients with heterozygous CTCF variants display dramatic neurodevelopmental phenotypes.

Variants in members of the core cohesin complex have also recently been implicated in human genetic syndromes, with considerable clinical overlap with CTCF deficiency. Germline heterozygous variants in RAD21 were first reported in 2012, when six patients were described [81]. Core features of DD, ID, and growth restriction appeared consistent with a milder presentation of Cornelia de Lange syndrome (CdLS), a severe neurodevelopmental disorder associated with DD, ID, growth restriction, and limb and other congenital anomalies, most commonly caused by variants in *NIPBL*, the protein responsible for loading cohesin onto DNA. Congenital anomalies, including cleft palate and congenital heart disease, were noted in this original report as well, consistent with the wide variety of congenital anomalies seen in CTCF deficiency, however, once again, these phenotypes were reported at a lower frequency than neurodevelopmental involvement. Over the past decade, there have been multiple additional reports of patients with a heterozygous germline variant in RAD21, which has strengthened the core features and broadened the phenotypic spectrum [82,83,84,85,86,87,88,89,90,91,92,93]. Recently, Krab et al. published a large case series describing 49 individuals with a RAD21 germline variant, which further solidified the centrality of DD and ID to the phenotype with less frequent involvement of other body systems. Similar to CTCF deficiency, the spectrum of neurocognitive involvement spanned from severe to minimal, including parents of probands with minimal known features [94]. Germline variants in SMC1A, SMC3, STAG1, and STAG2 have been identified as well over the past several years, with considerable phenotypic overlap with RAD21 deficiency, generalizing the phenotype of cohesin depletion. Variants in *SMC1A* and *SMC3*, respectively, have been identified in patients with neurocognitive impairment, growth restriction, and congenital anomalies [84,95,96,97,98,99,100,101,102,103,104,105,106]. Congenital anomalies are again less prevalent than neurocognitive effects, but variants in *SMC3* in particular seem to predispose to cardiac involvement. Variants in STAG1 were identified in six members of a large family in 2017 with autosomal dominant ID, and further analysis subsequently identified 11 additional unrelated patients with similar phenotypes [107]. In addition to the shared feature of ID, phenotypes found in a subset of the cohort also included microcephaly, epilepsy, growth restriction, autistic features, and congenital anomalies. Finally, STAG2 deficiency, termed Mullegama-Klein-Martinez syndrome, has been identified in several individuals over the past five years, with a consensus phenotype that also includes DD, ID, microcephaly, dysmorphic features, seizures, and a variety of congenital anomalies [93,108,109,110,111,112,113]. Similar to CTCF deficiency, advancements in the ability to perform unbiased sequencing on large cohorts of undiagnosed patients continues to identify an ever-increasing number of patients with germline cohesin-complex variants, including patients with intellectual disability, autism spectrum disorder, epilepsy, and holoprosencephaly [90,91,114,115,116].

**Figure 4 genes-13-00583-f004:**
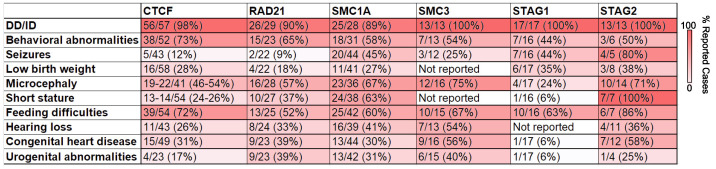
Prevalence of reported human phenotypes for patients with variants in CTCF or cohesin subunits. Color intensity represents the percentage of reported cases. Numbers are gathered from publicly available information from the following sources: [71,72] for CTCF; [94] for RAD21; [100] for SMC1A; [97] for SMC3; [107] for STAG1; [112] for STAG2.

In total, genetic syndromes have now been described for deficiency of CTCF and each member of the core cohesin complex, RAD21, SMC1A, SMC3, STAG1, and STAG2. Each is associated with a predominant neurodevelopmental phenotype, although involvement in other body systems to a lesser degree is also seen (Figure 5). Interestingly, although the role of somatic mutations in cohesin are becoming increasingly recognized in cancer, an increased risk of cancer in the patients described above with a germline variant has not been described (Reviewed in [117]). It’s important to note that cohesin and CTCF are involved in DNA repair [2,3], thus genome instability may lead to some of these described effects, particularly the cancer phenotypes. It is also noteworthy that with the exception of hemizygous males affected by presumably less damaging missense variants in the X-linked conditions associated with SMC1A and STAG2 deficiency, each of these disorders is characterized by heterozygous mutation, and that biallelic variants have not been described. This is in agreement with the embryonic lethality of homozygous depletion demonstrated in mice. Therefore, future studies should question why chromatin organization and gene expression are resistant to c.a. 50% depletion levels in cell lines, while heterozygote variants of CTCF and cohesin produce such pronounced phenotypes in patients.

## 3. Discussion

The human genetic conditions associated with CTCF and cohesin insufficiencies have made clear their necessity in neurodevelopment. It’s presumed that this necessity is due to their key role in forming CTCF loops, although the comparison to *Drosophila* outlined above suggests that CTCF’s role in gene expression may be distinct from its role in chromatin looping [29,46]. The phenotypes associated with CTCF and cohesin variants will undoubtedly expand as clinical genetic sequencing becomes more common, but currently the core features of developmental delay and intellectual disability are clearly shared in these conditions. Curiously, even though loss of CTCF and cohesin are thought to impact genome organization differently (Figure 1), there is a high degree of overlap between the patient phenotypes (Figure 4). Investigating sub-phenotypes specific to each may therefore be valuable and reflect the different mechanistic roles of CTCF vs. cohesin.

Animal models have been valuable tools to avoid some of the confounding factors inherent to patient studies. Studies in mice have reinforced the central role that CTCF and cohesin play in normal embryonic development, and tissue-specific depletion studies have highlighted the importance on neurodevelopment in particular. The embryonic lethality of animals with complete CTCF or cohesin deficiency has clearly illustrated the dose-dependent nature of phenotypes resulting from depletion of either gene product [51,52,53,54,55,56]. This point parallels the human genetic conditions in that there have not been reports of homozygously affected individuals in the literature, with the presumption that biallelic disease is likely embryonic lethal in humans as well.

The development of Hi-C [118] and additional techniques have allowed for the investigation into the molecular mechanisms underlying the functional phenotypes seen in human patients and animal studies. An interesting observation from these cell line models has been the relatively strong resistance to changes in genomic architecture after depletion of CTCF or cohesin, with the requirement for near complete depletion before significant loss of loops were noted [18,19,30,31,32,34,35,37,38,39,40,41]. This has important implications for the design of future experiments that model human genetic syndromes. All patients to date have presented with heterozygous variants (with some hemizygous examples of specific cohesin members), and mouse models have indicated that homozygous depletion is lethal, therefore how we reconcile the necessity for complete depletion in cell lines to that of the common in vivo state in human disease is an important consideration moving forward. Future models relying on heterozygous deletion, or introduction of specific variants, even with their presumed minimal effects on chromatin architecture, might therefore be good models for those seeking to understand associated human genetic syndromes [119].

Examination of the chromatin architecture in patient samples, particularly of neural tissue, would be important to confirm that patient-specific variants disrupt chromatin organization, but these have not to our knowledge been reported at this time. In the cell line models with incomplete depletion, changes to chromatin folding were modest but present, suggesting the presence of a small proportion of CTCF loops that were more sensitive than others to architectural protein dosage [18,30,31,32]. It is intriguing to speculate that these sites with the greatest dosage sensitivity, rather than those that appear more resistant to depletion of CTCF or cohesin, might prove to be of central importance in unraveling the effects on human neurodevelopment. This would therefore have important implications for future design of molecular experiments modeling the human conditions.

A second interesting observation from the clinical patients that highlights the importance of considering dosage levels in future experimental design, is that effects on neurodevelopment are present to at least some degree in nearly all patients identified thus far, while congenital anomalies of additional organs affect only smaller subsets of patients (Figure 4). The recurrence of congenital anomalies in multiple organ systems implies a dependence of CTCF loops in organogenesis broadly. Additional evidence for this is found in the small number of case reports identifying genetic variants in individual CTCF binding sites, instead of the CTCF gene itself. Disruption of these binding sites have been demonstrated to result in changes such as limb development or eye problems [22,23,24,25]. However, the lower frequency of these associated anomalies when compared to developmental effects in patients with CTCF or cohesin deficiency suggests that brain development in particular may be more sensitive to dosage changes of CTCF or cohesin than development of other organs, although this currently remains speculative. It is possible that ascertainment bias may be playing a role in this phenomenon, in that exome-scale genetic testing has only recently become widely available in the clinical setting, and the most severely affected patients, often with intellectual disability, have been more likely to have received genetic testing to this point. It does seem likely however that even as additional individuals are sequenced and the phenotypes broaden, effects on neurodevelopment may remain more prevalent than other organ involvement, and may reflect an underlying tissue-specific differences in sensitivity to depletion of CTCF and cohesin. Future work comparing the effects of depletion of these proteins on various tissues, either from clinical samples or organoid models, could provide exciting evidence toward this hypothesis.

In summary, studies of human patients, animal models, and cell lines have all demonstrated the critical roles for both CTCF and cohesin in neurogenesis, as well as a more conserved role in organogenesis broadly. Although technological advances allowing for near-complete depletion of these proteins have revealed striking differences in the effects on chromatin organization when compared to cell-line models with more modest levels of depletion, such drastic depletion is not tolerated in vivo in animal studies or human patients. The differences between the effects on chromatin organization in the partially depleted versus the completely depleted states may therefore be revealing of particularly sensitive architectural features that may play an outsized role in the phenotype of the human genetic syndromes. Additionally, striking differences in prevalence between the involvement of neurodevelopmental phenotypes and other body system involvement in these clinical syndromes may reflect an underlying tissue-specific differential in sensitivity to dosage alterations of CTCF and cohesin, and is worthy of further exploration. Taken together, future inquiry into each of these lines of evidence is important going forward in order to more fully uncover the mechanisms underlying these important human neurodevelopmental disorders, as the first step toward the development of effective interventions.

## Figures and Tables

**Figure 1 genes-13-00583-f001:**
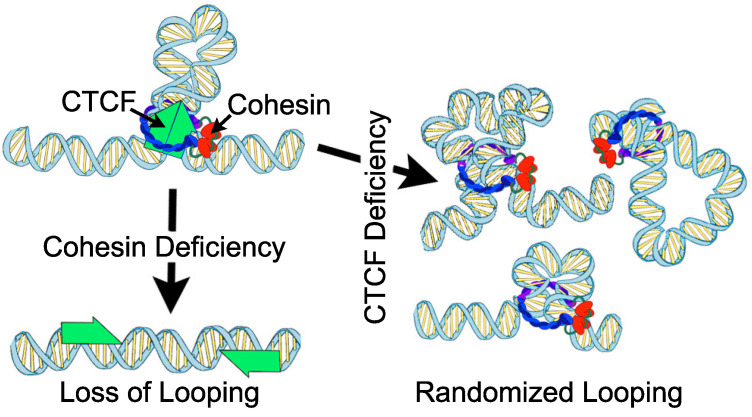
Different outcomes of CTCF v.s. cohesin deficiencies. CTCF and cohesin work in concert to establish loops between CTCF anchors. Loss of either results in a loss of CTCF-specific loops. However, due to their distinct roles, the loss of CTCF is thought to have different molecular impacts from the loss of cohesin.

**Figure 2 genes-13-00583-f002:**
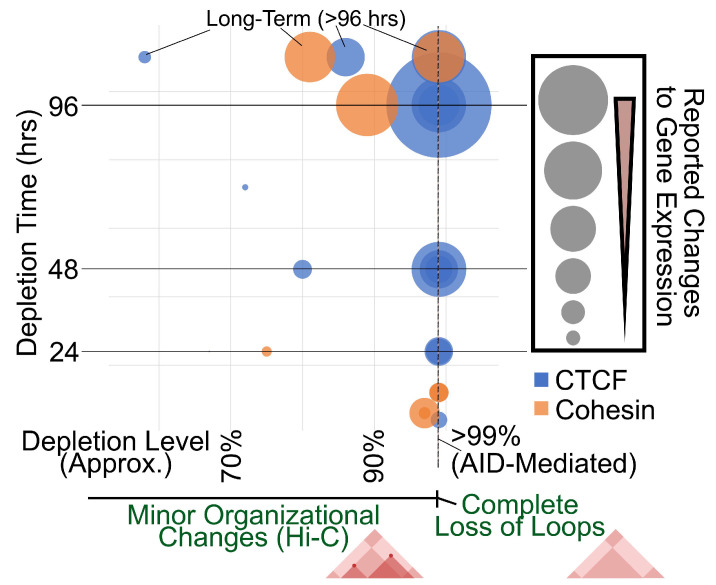
Gene Expression Defects in Response to Dosage and Time of depleting CTCF (blue) or cohesin (orange). Size of the circles represent the number of differential genes compiled from publicly available data in several reports [18,19,30,31,32,34,35,37,38,39,40,41]. Depletion levels in these reports were often approximated or the midpoint was plotted if a range was given. AID studies often report “complete depletion” and thus were plotted as “>99%”. Depletion times larger than 96 h were plotted as “long-term”. Concentric circles indicate different studies that used similar depletion time and level, but reported different numbers of differential genes.

**Figure 3 genes-13-00583-f003:**
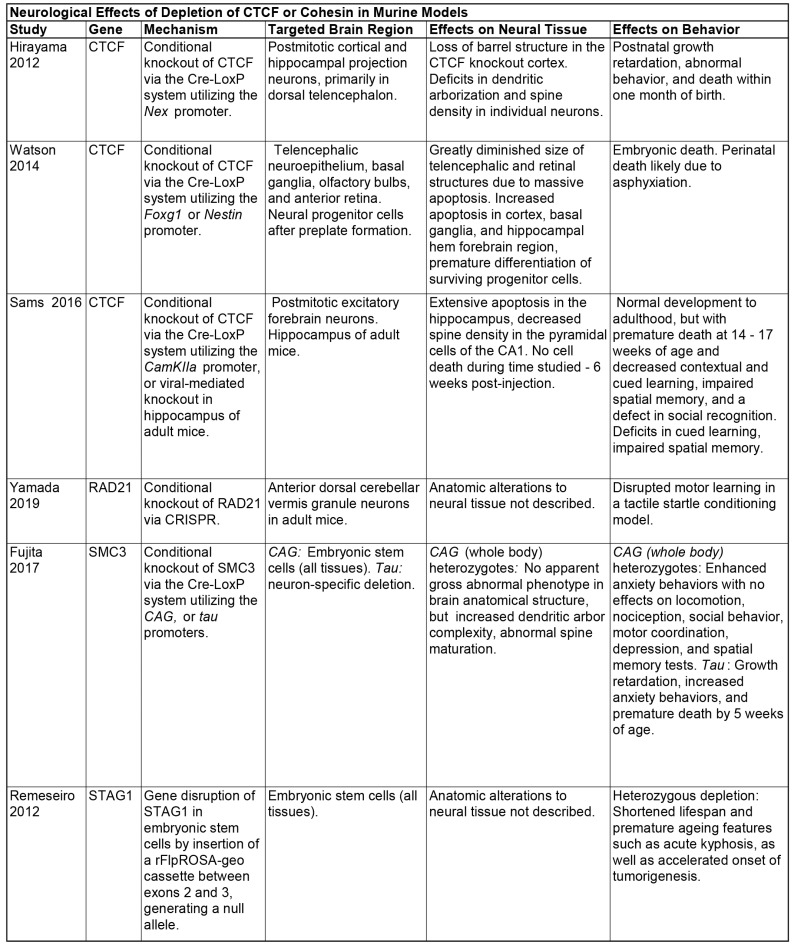
Effects of CTCF and Cohesin Depletion in Murine Models. Summary of effects compiled from publicly available information found in [54,55,57,58,59,60].

**Figure 5 genes-13-00583-f005:**
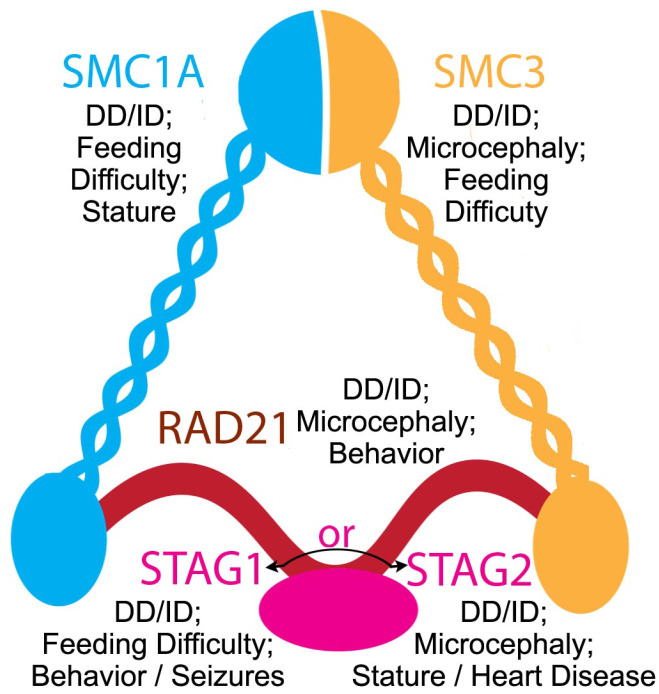
Cohesin subunits and the most frequently reported phenotypes associated with mutations in patients, as described above.

## Data Availability

Not applicable.

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
