# Peer review of "Implications of Dosage Deficiencies in CTCF and Cohesin on Genome Organization, Gene Expression, and Human Neurodevelopment"

_genes, 2022, doi:10.3390/genes13040583_

Round 1

Reviewer 1 Report

This is a well-written review on CTCF and cohesin on chromatin organization and gene regulation, with a focus on human neurodevelopment. The authors first reviewed the recent discoveries of dosage tolerance of CTCF and cohesin on chromatin organization and gene expression in cell line-based CTCF or Cohesin Knockdown or KO studies. Only near completion deletion of CTCF and Cohesin lead to complete loss of chromatin loops but limited impact on gene expression. Then they discussed in vivo evidence of Dosage sensitivity of CTCF and cohesin, both from mouse models and neurodevelopment-related patients with heterozygous germline variants of CTCF or Cohesin. I found the review to be comprehensive and objective with the focus on neurodevelopment. It could be a valuable resource to the field. Below are some comments to the authors

Major:

  • Reference are missing
  • Chromatin loops could be formed by mechanisms, such as promoter and enhancer interactions, other than CTCF and cohesion. It is inaccurate to say “A second organizational layer is a multitude of high-intensity chromatin loops formed by CCTC-Binding Factor (CTCF) and cohesion”
  • Line 120-125 is solely relying on the loop-extrusion model, which is the most popular model in the field. However, not all the loop anchors are associated with cohesin and this model also has its limitations. For example, this model relies on active translocation of cohesion along chromatin but there is no evidence to support the active translocation capability of cohesion. This model does not explain how cohesion complex successfully slide through some CTCF sites while stops at certain sites. The limitation of this model and acknowledgment of other models should be clearly stated to avoid confusion.
  • It is possible that dosage of CTCF and cohesin complex is critical for the proper regulation of a limited number of genes essential for neurodevelopment, rather than the global changes. This is consistent with the variant phenotype being mostly restricted to neurodevelopment. In addition, Genome-wide assays, like HiC, lack the resolution of confidently capturing spatiotemporally regulated dynamic loops. The authors could better highlight the limitations of such assays used in evaluating the knockdown experiments.
  • Confounding factors can not be ruled out from the patient studies, calls for animal models

Minor:

  • The color of cohesion loops and DNA double helix could be adjusted to allow better visualization of the structures. The current colors in figure 1 make it difficult to see the cohesion loops.

  • Line 118-119, another possible explanation of the long depletion times correlated with more dramatic effects is the secondary effects from the first round of gene expression changes directly influenced by CTCF loop disruption.
  • Page 161-168: The difference between mammalian and Drosophila CTCF should be pointed out. In drosophila, CTCF is known to work with CP190 instead of cohesion. So the mechanisms might be different.

Author Response

We thank the reviewers for the kind comments and helpful critiques. We have strived to implement and address all reviewer comments and feel the manuscript has improved due to this process.

Reviewer 1

Comments and Suggestions for Authors

This is a well-written review on CTCF and cohesin on chromatin organization and gene regulation, with a focus on human neurodevelopment. The authors first reviewed the recent discoveries of dosage tolerance of CTCF and cohesin on chromatin organization and gene expression in cell line-based CTCF or Cohesin Knockdown or KO studies. Only near completion deletion of CTCF and Cohesin lead to complete loss of chromatin loops but limited impact on gene expression. Then they discussed in vivo evidence of Dosage sensitivity of CTCF and cohesin, both from mouse models and neurodevelopment-related patients with heterozygous germline variants of CTCF or Cohesin. I found the review to be comprehensive and objective with the focus on neurodevelopment. It could be a valuable resource to the field. Below are some comments to the authors

Major:

  • Reference are missing

We apologize for this error in LaTex formatting and have corrected this for the updated draft.

  • Chromatin loops could be formed by mechanisms, such as promoter and enhancer interactions, other than CTCF and cohesion. It is inaccurate to say “A second organizational layer is a multitude of high-intensity chromatin loops formed by CCTC-Binding Factor (CTCF) and cohesion”

We have modified this statement such that chromatin organization is not described as two layers (compartmentalization and CTCF looping), and rather introduce CTCF looping as one of multiple interconnected organizational layers, in a more inclusive manner as suggested by the Reviewer.

(Lines 27 – 30)  “A prominent layer of chromatin organization is the segregation of active and inactive chromatin into A and B compartments. Another layer are chromatin loops forming enhancer-promoter interactions. As part of these multiple interconnected organizational layer are the multitude of high-intensity chromatin loops characterized by CCTC-Binding Factor (CTCF) and cohesin.”

  • Line 120-125 is solely relying on the loop-extrusion model, which is the most popular model in the field. However, not all the loop anchors are associated with cohesin and this model also has its limitations. For example, this model relies on active translocation of cohesion along chromatin but there is no evidence to support the active translocation capability of cohesion. This model does not explain how cohesion complex successfully slide through some CTCF sites while stops at certain sites. The limitation of this model and acknowledgment of other models should be clearly stated to avoid confusion.

We agree with these important points regarding the limits of the loops extrusion model, and have added language addressing this.

(Lines 40-42) “In recent years, the loop extrusion model has been supported by a large amount of evidence, although limitations to the model remain, including an explanation for the ability of cohesin to bypass some CTCF cites while passing others”

  • It is possible that dosage of CTCF and cohesin complex is critical for the proper regulation of a limited number of genes essential for neurodevelopment, rather than the global changes. This is consistent with the variant phenotype being mostly restricted to neurodevelopment. In addition, Genome-wide assays, like HiC, lack the resolution of confidently capturing spatiotemporally regulated dynamic loops. The authors could better highlight the limitations of such assays used in evaluating the knockdown experiments.

We agree with that there are important limitations to HiC, and also that spatiotemporal regulation of dynamic loops may play an important role in the underlying biology of the described phenotypes. We have added statements acknowledging this.

(Lines 157-160) “Related to this discrepancy, it is important to note the limitations of these cell-line depletion experiments assessing chromatin architecture, for example that techniques such as Hi-C typically do not have the regulation of CTCF loops very highly resolved in a spatiotemporal manner, which may play a role in the observed phenotypes.”

  • Confounding factors can not be ruled out from the patient studies, calls for animal models.

We agree with the importance of animal models to reduce the confounding factors mentioned in clinical studies and have added statements to support this.

(Lines 182-184) “Murine models have proven useful to evaluate the phenotypic impacts of the loss of mammalian CTCF and cohesin \textit{in vivo}, and are a critical tool for future research given the reduction of confounding factors when compared to studies focused solely on clinical patients.”

(Lines 313-314) “Animal models have been valuable tools to avoid some of the confounding factors inherent to patient studies.”

Minor:

  • The color of cohesion loops and DNA double helix could be adjusted to allow better visualization of the structures. The current colors in figure 1 make it difficult to see the cohesion loops.

We agree and have adjusted the coloring to increase contrast.

  • Line 118-119, another possible explanation of the long depletion times correlated with more dramatic effects is the secondary effects from the first round of gene expression changes directly influenced by CTCF loop disruption.

We agree with this interpretation and have added a statement directly stating this.

(Lines 123-125) “Additionally, it is possible that the immediate changes in the expression of a small number of genes lead to secondary effects that culminate in the larger expression changes seen at longer time-points.”

  • Page 161-168: The difference between mammalian and Drosophila CTCF should be pointed out. In drosophila, CTCF is known to work with CP190 instead of cohesion. So the mechanisms might be different.

We agree this is an important distinction between these two species and have added a statement addressing this.

(Lines 176-178) “It is also important to note that \textit{Drosophila} CTCF interacts with, and binds chromatin alongside, other architectural proteins such as CP190, indicating that its mechanisms may function differently”

Reviewer 2 Report

Cummings et al described in this review the critical role of CTCF and Cohesin in human neurodevelopment. The manuscript is well organized and is considered an update. 

Nevertheless, it is very well documented the important interaction between CTCF, Cohesin, the formation of interchromosomal aberrations, CpG methylation and centromeres. The authors completely avoided these interactions. 

In addition, the role of DNA repair mechanisms in these pathologies associated with this deficiency must be better described (the role of Rad 21 and certainly other genes ).

The paragraph Main is very long and needs to be rewritten.   There is a problem in the references.

Author Response

We thank the reviewers for the kind comments and helpful critiques. We have strived to implement and address all reviewer comments and feel the manuscript has improved due to this process.

Reviewer 2

  • Cummings et al described in this review the critical role of CTCF and Cohesin in human neurodevelopment. The manuscript is well organized and is considered an update. Nevertheless, it is very well documented the important interaction between CTCF, Cohesin, the formation of interchromosomal aberrations, CpG methylation and centromeres. The authors completely avoided these interactions. 

We agree that these are important interactions and that CTCF and cohesin have been implicated in an ever-increasing number of important cellular functions. Although this review is centered on the impact on gene expression and neurological implications, it is important that these other relationships be highlighted. We include statements highlighting these relationships in the Introduction and turn the reader towards other review articles with more focus on these aspects.

(Lines 31-33) “CTCF and cohesin have been implicated in a multitude of diverse cellular functions, including DNA repair, sister chromatid cohesion, and the proper localization of DNA methylation and histone modifications (Reviewed in \cite{Phipps2022, Yatskevich2019})”

  • In addition, the role of DNA repair mechanisms in these pathologies associated with this deficiency must be better described (the role of Rad 21 and certainly other genes ).

Similarly, we agree that the role of cohesin in DNA repair is critically important, and although our review cannot fully focus on this aspect, we have referenced to other reviews that do (as above). We have also included a statement regarding the importance of considering the function in DNA repair when considering these phenotypes.

(Lines 287-292) “Interestingly, although the role of somatic mutations in cohesin are becoming increasingly recognized in cancer, an increased risk of cancer in the patients described above with a germline variant has not been described (Reviewed in \cite{Waldman2020}). It's important to note that cohesin and CTCF are involved in DNA repair\cite{Phipps2022, Yatskevich2019}, thus genome instability may lead to some of these described effects, particularly the cancer phenotypes.”

  • The paragraph Main is very long and needs to be rewritten.  

We appreciate the comment and have done our best to create manageable subheadings within the Main section.

  • There is a problem in the references.

We apologize for this error in LaTex formatting and have corrected this for the updated draft.

Round 2

Reviewer 1 Report

All my critiques have been addressed

Reviewer 2 Report

the manuscript can be accepted